# The effect of supply chain risks management practices on operational performance of pharmaceutical manufacturing companies in Addis Ababa, Ethiopia: Analytical cross-sectional study

Dinku Mechal[1], Bekele Boche [2]*

**1** Department of pharmacy, College of Medicine and Health Sciences, Wachamo University, Central Ethiopia Regional State, Hosaina, Ethiopia, **2** Department of Social and Administrative Pharmacy, School of Pharmacy, Faculty of Health Sciences, Jimma University, Oromia, Ethiopia

\* bekelebeke@gmail.com

## Abstract

### Background

The efficient movement of pharmaceuticals through various stakeholders to reach customers in the appropriate quantity and at the right time is achieved through supply chain management. However, the intricate nature of supply chain processes poses tremendous supply chain risks and jeopardizing pharmaceutical manufacturing company's ability. Failure to address these risks can hinder the provision of high-quality health services. However, effective supply chain risk management can foster more resilient and efficient global supply chains. Therefore, this study aimed to investigate the effect of pharmaceutical supply chain risks management practices on operational performance of pharmaceutical manufacturing companies in Addis Ababa, Ethiopia.

### Methods

Analytical cross-sectional study complemented with qualitative method was conducted at pharmaceutical manufacturing companies in Addis Ababa between May-August 2023. One hundred seventy two staffs working in four manufacturing companies included in the study. For quantitative part, pretested a self-administered five-point Likert scale questionnaire was used and analyzed using SPSS® -version 26. Assumptions of linear multivariate regression were checked and the level of significance determined at a 95% CI and p-value <0.05. Nine face to face in-depth interviews with key informants were conducted to gather the qualitative data, and the data were analyzed using thematic analysis technique.

**Data availability statement:** All relevant data are within the paper and its Supporting Information files.

**Funding:** The author(s) received no specific funding for this work.

**Competing interests:** The authors declare that they have no known competing financial interests or personal relationships that could have appeared to influence the work reported in this paper.

**Abbreviations:** DR: Demand risk; SR: Supply risk; PSC: Pharmaceutical supply chain; SCRM: Supply chain risk management; SPSS: Statistical package for the social sciences

## Result

The study included 172 employees from four manufacturing companies, with a response rate of 97%. The regression analysis revealed that demand ($\beta$=-0.191, t = -4.162, $p < 0.05$) and supply side risks ($\beta$=-0.131, t = -2.015, $p < 0.05$) have a negative effect on the operational performance of manufacturing companies. Holding other variables constant, a one-unit increase in demand and supply side risks results in 19.1% and 13.1% lead to decrease in operational performance of manufacturing companies, respectively. Increasing costs of freight, shortage and fluctuating foreign exchange rate for currency, lack of logistics expertise and organized risks mitigation team were the major challenges for manufacturing company's operational performance.

## Conclusion

Demand and supply side risks affected the supply chain performance of the manufacturing companies. Furthermore, the increasing costs of freight, shortage of foreign currency, lack of logistics expertise and organized risk mitigation team were the main bottleneck for pharmaceutical manufacturing company's performance. The study result suggests demand and supply side risks, increasing costs of freight; shortage foreign currency exchange rate and lack of logistics expertise in companies should be given attention by stakeholders to improve operational performance. Furthermore, understanding these risk factors can improve the operational performance of the pharmaceutical industry by influencing policy and industry practices in the larger framework of global supply chain risk management not only one country.

## Introduction

Supply chain management is an integrating role primarily responsible for connecting business units and processes inside and between firms to create a coherent and high-performing business model. It encompasses logistics management tasks such as selection, need anticipation, manufacturing operations, inventory management, drives process, and activity coordination with manufacturers, suppliers, and distributors working across marketing, sales, product design, finance, and information technology [1,2]. To be compatible with access to medication which is the pillar of the right to health, the pharmaceutical supply chain (PSC) should offer medications in the correct amount, with acceptable quality, to the right place and consumers, at the right time, and at the lowest possible cost [3,4]. In these processes, pharmaceutical manufacturing companies play an imperative role in producing and delivering medications and protecting human life [5].

However, in fulfilling responsibilities, the pharmaceutical plant may face many supply chain problems, including raw material shortages, product quality concerns, short product life cycles, workers' skills gaps and commitment, seasonal product demand, high expiration, and damage, leak out of the pipeline, human-made and

natural disasters and cold chain management difficulties resulted in quality problems and financial burdens. Any threat to the PSC might affect the health system's efficiency and interrupt pharmaceutical delivery to customers. To address such vulnerabilities, it is necessary to identify the associated supply chain risks and take steps to mitigate them in order to assure the best practices in the PSC for medicine quality ingredients and business flexibility [5,6].

Supply chain risks are any incident or event that could disrupt the movement and flow of materials across the chain. It is the negative variance from the anticipated value of those performance metrics, which leads to undesirable consequences for the firm and it also equated to supply chain disruption. So, supply chain risks have become a significant concern for many businesses in today's unpredictable and turbulent markets [6,7]. Therefore, pharmaceutical manufacturing firms should give due attention to supply chain risk management (SCRM) practices, where it is the systematic identification, appraisal, and mitigation of potential disruptions in logistics networks with the intention of minimizing their negative influence on the logistics network's performance [8]. To establish and implement successful supply chain risk mitigation strategies, companies and supply chain participants must first understand and identify supply chain risks, as well as the events and conditions that produce them. Early identification and awareness of risks can assist pharmaceutical manufacturing companies in avoiding or reducing unnecessary costs, liabilities, waste, and increasing efficiency of the supply chain [5,9].

Pharmaceutical manufacturing companies should handle the supply chain risks through collaboration with key suppliers, distributors and consumers, sharing knowledge, information and resources among the partners, keeping buffer stocks, develop and uphold contingency plans (contain basic guidance, direction, transparency, responsibilities, and administrative information), transferring and minimizing risks through insurance and a well-established quality control process. Minimizing the supply chain threats through the above means will help to increase firm operational performance to achieve its objectives [10,11]. Operational performance refers to the comprehensive supply chain operations leveraged to achieve customer anticipations such as product availability, optimum cost, service quality, delivery time and reliability, and flexibility in providing services [12].

Hence, pharmaceutical manufacturing companies should measure and optimize the quality of services they provided, flexibility, services delivery and cost minimization to be profitable and meet customer requirements. On the other hand overlooked supply chain risk management in pharmaceutical sector imposes paramount challenges on manufacturing firm performance. For instance, a study conducted in Pakistan on supply chain risk management and operational performance, it was found that risk management practices had a significant impact on operational performance ($\beta=0.481$, $p < 0.001$). This result highlights the importance of prioritizing risk management practices to improve operational performance [13]. Moreover, one of the systematic reviews conducted in Iran on the pharmaceutical supply chain risks revealed that approximately 40% of the risks are attributed to the supplier side, while 28% are related to regulatory risks. [14].

Supply chain risk management extends beyond addressing immediate issues; it necessitates a continuous commitment from stakeholders and participants to minimize and mitigates the negative consequences of those risks. Organizations should maintain preparedness and risk response strategies as essential prerequisites. For instance, research conducted among pharmaceutical companies in Vietnam revealed that the supply side risks and demand side risks had a significantly negative impact on company performance, with coefficients of 0.480 and 0.534, respectively. Additionally, supply chain risks exhibit a negative influence on companies' operational performance of supply chain integration with an impact factor of 0.360, significant at the 1% level (P-value = 0.000) [15,16].

The pharmaceutical supply sectors in most African countries are facing significant challenges that are affecting their performance. These challenges include stock-outs, counterfeit and fake products, disruptions, expired drugs, corruption, weak regulatory systems, inadequate infrastructure, hijacking, insufficient information and communication systems, inadequate storage facilities, and a lack of efficient management procedures and long lead times. Moreover, poor

pharmaceutical products the quantification, a lack of transparent procurement procedures, inadequate planning, monitoring, evaluation, and insufficient budget allocation are challenging [16–18]. Also, according to a study of the Nigerian pharmaceutical supply chain, the main obstacles affecting supply organization performance are excessive product stocking, unpredictable supplier price increases, and failure to deliver items to consumers, supplier product quality issues, product expiry, and a lack of funds [19].

Studies conducted in Moroccan and Indian pharmaceutical companies found that medication manufacturing firms face significant risks, including forecast inaccuracy, unexpected demand changes, delivery chain interruptions, declining market prices, unpredictability of trade restrictions, transportation failure, supplier failure, and delayed supplier delivery. These risks can have a major impact on the performance of the pharmaceutical companies [20,21].

Ethiopia has emerged as one of Africa's fastest-growing economies with the low performance of the manufacturing sector, which has been unimpressive and observed below from becoming a driver of development and economic change. It accounts only for 6.4% of the growth development program (GDP) in 2017. Pharmaceutical firms are one of manufacturing companies in Ethiopia, where most of them operate at or near capacity and supply only 20% of the local market. They face several challenges, including supply chain disruptions, counterfeits, and illegal pharmaceutical trade, a lack of foreign currency, a heavy reliance on imports for manufacturing inputs, insufficient regulatory capacity, a lack of skilled and committed human resources, financial, technological, legal, and weak firm relations. This led to high unmet demand for essential medications in the country [22,23].

For example, survey conducted at Ethiopian pharmaceutical supply services indicated there were forecasting inaccuracy, unexpected surge in demand, currency fluctuation, supplier risk (single supplier, longer supplier lead-time and item unavailability), quality assurance-related, rapid employee turnover, lack of sufficient fund, expired inventory, lack of sufficient skilled human resources [24].

Despite the study conduct at Ethiopian pharmaceutical supply services which revealed that inadequate supply risk management, as far as our knowledge is concerned, there are limited studies globally and none conducted in Ethiopian on pharmaceutical manufacturing companies supply chain risk management. Therefore, this study aimed to assess effect of supply chain risks management practices on operational performance of pharmaceutical manufacturing companies in Addis Ababa, Ethiopia.

The study theoretical framework was presented in Fig 1.

The proposed hypotheses are the following

H1: Demand side risk negatively affects operational performance

H2: Supply side risk negatively affects operational performance

H3: Catastrophic side risk negatively affects operational performance

H4. Production side risk negatively affects operational performance

H5: Regulatory side risk negatively affects operational performance

H6: Financial side risk negatively affects operational performance

H7: Infrastructure side risk negatively affects operational performance

Concerning the hypotheses mentioned above, H1, H4, and H6 find support in studies conducted in India on the development and validation of the supply chain risk scale, as well as on the assessment of supply chain risk factors in the pharmaceutical industry for performance enhancement [25,26]. On the other hand, H2, H3, H5, and H7 are supported by a study conducted in Addis Ababa, Ethiopia [27]. Also, H1, H2, and H6 are backed by a study on decision modeling of risks in pharmaceutical supply chains in Bangladesh [5].

**Theoretical framework and hypotheses**

**Exogenous (Independent) variables**

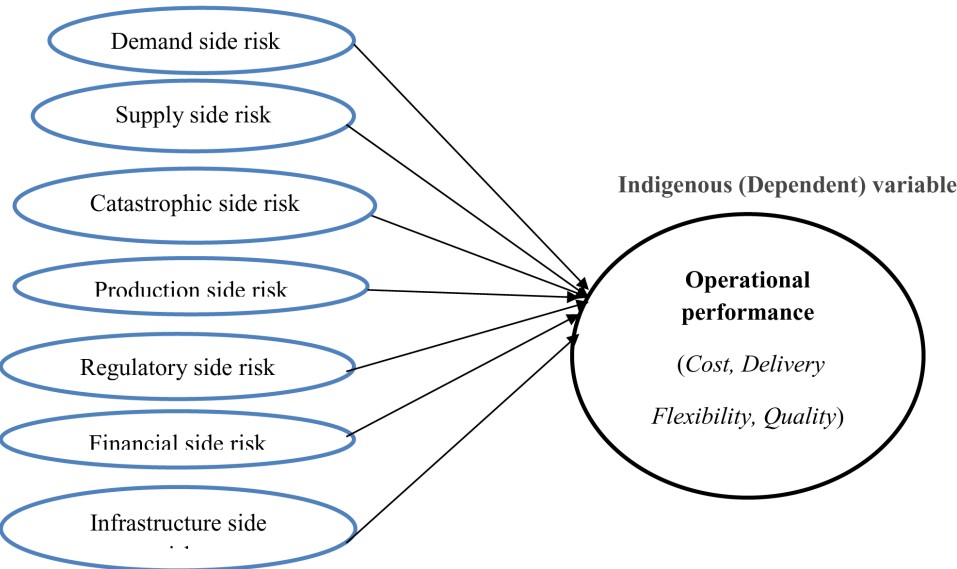

**Fig 1. Theoretical framework.**

A 2019 study on India's pharmaceutical industry found that supply-side risks, production-side risks, and infrastructure-side risks all negatively impact operational performance (OP). Supply-side risks decrease OP ($\beta = 0.176$, t = 2.678, p < 0.01), production-side risks lower OP ($\beta = 0.602$, t = 7.450, p < 0.01), and infrastructure-side risks reduce OP ($\beta = 0.344$, t = 3.033, p < 0.01). This supported the hypotheses (H2, H4, H6, and H7) [28]. Again, the review of pharmaceutical supply chain risks identified that supplier-related risks were the most critical; representing 40% of all risks impacting OP. Regulatory risks were also widespread, impacting the pharmaceutical supply chain's OP. Additionally, financial risks such as tax variations, supply expenses, interest rate fluctuations, and tariff policy changes are other factors influencing overall OP [14]. So, this assessment supports H2, H5 and H6. Moreover, the similar study conducted in Iran highlighted that financial and regulatory risks are the top pharmaceutical supply chain risks that affect the operational performance of organizations [29].

## Methods

### Study area, design, and period

An analytical cross-sectional study complemented by a qualitative approach was conducted at the pharmaceutical manufacturing companies located in Addis Ababa, Ethiopia, namely Ethiopian Pharmaceutical Manufacturing (EPHARM), East Africa Pharmaceutical Manufacturing (EAP), Julphar Pharmaceutical Manufacturing (JPM) and Addis pharmaceutical manufacturing between May to August, 2023. The total staffs of pharmaceutical manufacturing companies are 764; where Ethiopian Pharmaceutical Manufacturing 360, East Africa Pharmaceutical Manufacturing 144, Julphar Pharmaceutical Manufacturing 140 and Addis pharmaceutical manufacturing were contain 120 staffs.

## Population and sampling procedure

The study focused on pharmaceutical companies found in Addis Ababa and engaged in direct manufacturing and their staff as the source population. The study population consisted of employees responsible for pharmaceutical supply chain management (PSC) operations at pharmaceutical manufacturing firms. Those pharmaceutical companies participated in pilot study, workers who refuse to take part and not present at the time of data collection were excluded from the study. Ethiopia has 11 pharmaceutical manufacturing companies out of which most of them found in Addis Ababa and Tigray region. However, due to the instability and security issues in Tigray region over the past year, it was excluded from the study. In Addis Ababa there are five pharmaceutical companies which directly engaged in manufacturing. Out of those four was included in the study and the rest in the pilot study. One hundred seventy-seven professionals were directly involved on PSC operations at those pharmaceutical manufacturing companies in Addis Ababa. Since this figure was manageable and easy to address, we considered all of them in the study. Concerning the qualitative assessment, we conducted in depth face-to-face interviews with nine key informants (KIs) chosen purposively based on the position and service experience they had in their company and sample size was determined by saturation of ideas principles.

## Ethics Approval

The study was conducted in accordance with the declaration of Helsinki. Prior to commencing the actual data collection, the study obtained ethical clearance and approval from the Institutional Review Board of Jimma University, Institute of Health, with reference number JUIH/RB/5/22 on June 27, 2022.

## Consent to Participate

The study ensured that participation was entirely voluntary and confidential, with measures in place to protect the privacy of participants and their sensitive information. Each participant was provided with a written informed consent form before data collection, and their rights to withdraw or decline participation were fully respected. Only participants who provided their explicit consent were included in the survey.

## Data collection procedures

Data was collected by utilizing structured self-administered questionnaires and interview guides, which were developed after conducting a thorough review of relevant literature including books and journal articles [30–32]. The self-administered questionnaires consisted of four parts. Part I solicited socio-demographic information from respondents through five questions. Part II comprised 32 questions focused on supply chain risks. Part III included 16 questions pertaining to supply chain risk mitigation practices. Part IV addressed operational performance and contained 11 questions. Parts II–IV utilized agreement-type questions on a five-point Likert scale, ranging from '1' indicating 'strongly disagree' to '5' indicating 'strongly agree' (*Supplementary file survey questioner annex*). The data were collected by three data collectors (druggists) who are familiar with the area and closely supervised by the investigators. Additionally, a qualitative section was included, consisting of five interview questions. Prior to the interviews, written informed consent was obtained from the participants, allowing for the audiotaping and note taking of the interviews. Then, face-to-face in-depth interviews were conducted by the principal investigator using local language, Amharic. During the interviews, participants were asked in-depth and probing questions to gather comprehensive information on the topic of interest and on average; each interview lasted for approximately 25 minutes.

## Data processing and analysis

The quantitative data underwent sorting and coding before being entered into Epidata and subsequently into SPSS version 26 software for analysis. Descriptive and inferential statistical analyses were conducted to summarize the findings.

Descriptive analyses of the data were performed using SPSS and presented the results in terms of frequency, mean, and percentages. For the inferential statistical analysis, multiple linear regressions were employed. To ensure the validity of the linear regressions, assumptions such as normality, linearity, multicollinearity, and correlations were assessed prior to the final analyses. The analysis output indicated no violation of the underlying assumptions of multivariate linear regressions analysis. Normality was evaluated using measures such as histograms, skewness, and kurtosis, with an absolute Z value of less than 3.29 considered acceptable for sample sizes less than 300. Linearity was assessed by inspecting scatter plots of residuals, with residuals falling on the diagonal regression line indicating linearity. Multicollinearity among independent variables was checked using tolerance and VIF (Variance Inflation Factor), with cut-off points of >0.2 and <3, respectively. Pearson's product-moment correlation coefficient (r) was used to quantify the strength of linear relationships between variables, with values ranging from -1 to 1. Rule of thumb guidelines were used to interpret correlation values, with ($0.1 < r < 0.3$) indicating weak correlation, ($0.4 < r < 0.6$) indicating moderate correlation, and ($0.7 < r < 0.9$) indicating strong correlation [33,34]. The inferential statistical analysis was conducted to determine associations between dependent and independent variables, with a p-value < 0.05 considered significant. The analyzed data were presented using frequency and percentage tables.

The qualitative data were analysed manually using a thematic analysis technique. First, the investigators and one of English language expert staff at Jimma University, school of English transcribed and translated the recordings into the English language. A qualitative research expert at Jimma University verified the accuracy of the transcription and translation. Then variables were coded and arranged in a word document in a tabular form, and themes were identified by combining the variables with similar codes. These include Cost/currency related issues, Natural and man-made threats, Logistics related issues, and supply chain risk management methods and constraints. Finally, the themes were described in narration form, followed by the quotations of some opinions.

## Data quality assurance and Trustworthiness

A pre-test was conducted before the main study to assess the understandability and applicability of the instrument at Pharmacure pharmaceutical manufacturing company in Addis Ababa, Ethiopia. The pre-test included 18 participants, constituting 10% of the main sample size, and encompassed 71 questions. The completeness and consistency of the filled instruments were manually checked. Then, the reliability test was performed, and adjustments were made to the questions based on the Cronbach's alpha values. Consequently, five out of the 71 questions administered during the pre-test were found to be unreliable and were removed. After deleting these questions, the Cronbach's alpha coefficients for each construct (pharmaceutical supply chain risks, pharmaceutical supply chain risk mitigation practices, and operational performance) ranged from 0.787 to 0.908, which falls within the acceptable range (α > 0.53) [35]. Three data collectors with a background in pharmacy were recruited to collect the quantitative data under the supervision of the principal investigator on a daily basis. The principal investigator provided training to the data collectors, explaining the study's objectives, relevance, protocol, and details regarding participation.

In-depth interviews were conducted with the investigators to maintain consistency in the information gathered and its analysis. Following data collection, a clear coding scheme was developed before analyzing the text. This ensured consistency in how different segments of text were categorized and analyzed. The methodological descriptions guaranteed the study's transferability and dependability, while an explanation of the whole research process further enhanced the trustworthiness of the results. Thematic analysis techniques were then applied to identify recurring themes and patterns in the text data, aiding in organizing and interpreting qualitative data effectively. Multiple data sources or methods were used to verify findings from text analysis, enhancing the credibility and robustness of the analysis. Detailed documentation of the analysis process, including coding decisions, interpretations, and any changes made during the analysis, was maintained. This helped ensure transparency and reproducibility.

# Result

## Demographic profile of the study respondents

This investigation aims to evaluate the effect of supply chain risks management practices on operational performance of pharmaceutical manufacturing companies in Addis Ababa, Ethiopia. A total of 177 questionnaires were distributed to respondents, out of which 175 (98.3%) were received back. After removing three outliers, 172 (97%) questionnaires were found to be complete and suitable for analysis. Among the respondents, the majority were male, accounting for 77.3%. Regarding education level, 122 respondents (70.9%) held a bachelor's degree, while 50 respondents (29.1%) had a master's degree. The study findings revealed that out of the 172 respondents, 71(41.28%) had less than 5 years of work experience and the majority of respondents, 134 (77.91%), were employed in the supply chain management department (Table 1).

## Supply chain risks of pharmaceutical manufacturing companies

Among the 172 respondents, 84 (48.8%) agreed that the company faces unanticipated or volatile customer demand, with a mean score of 3.57 (SD=1.038) and 79 respondents (45.9%) agreed that there is distorted information regarding demand and supply of customers, with a mean score of 3.59 (SD=1.148). In terms of demand-side risks in the pharmaceutical supply chain, the entire constructs received mean scores greater than 3.5. Regarding supply-side risks, 87 respondents (50.6%) agreed that there are deprived logistic service performances from suppliers, with a mean score of 3.73 (SD=0.972) and 82 respondents (47.7%) agreed that the company is dependent only on its key suppliers, with a mean score of 3.43 (SD=1.026). The majority of respondents with mean score above 3.41 agreed with the constructs related to supply-side risks.

Regarding production-side risks, 84 respondents (48.8%) disagreed with regard to low product quality and safety-related risks in the company, with a mean score of 2.48 (SD=0.994). Furthermore, 76 respondents (44.2%) disagreed with the risk related to the lack of skilled workers in the company, with a mean score of 2.67 (SD=1.103). For the infrastructure risks, 65 respondents (37.8%) agreed that there is a breakdown of internal IT, with a mean score of 3.09 (SD=1.174) and also, 50 respondents (29.1%) agreed with a mean score of 3.02 that there is a breakdown of external IT infrastructure. In terms of financial risks, 63 respondents (36.6%) agreed that there is a risk related to dynamic foreign exchange rates,

**Table 1. Socio-demographic profile of the respondents in pharmaceutical company of Addis Ababa, Ethiopia, 2023 (N=172).**

| Characteristics (variables) | | Frequency (percent) |
|---|---|---|
| Gender | Male | 133(77.3) |
| | Female | 39 (22.7) |
| Age (by years) | <25 | 16(9.3) |
| | 25-40 | 132(76.75) |
| | 40 and above | 24(13.95) |
| Working department | Supply chain | 134(77.91) |
| | Research and development | 19(11.05) |
| | Quality control and quality assurance | 19(11.05) |
| Educational level | Degree | 122(70.9) |
| | Master and above | 50(29.1) |
| Year of experience | Below five | 71(41.28) |
| | 5-10 | 73(42.44) |
| | Above 10 | 28(16.28) |

with mean scores of 3.25 (SD=1.071). and 56 respondents (32.6%) agreed that there is a financial restriction-related risk, with a mean score of 3.24 (SD=1.092). The mean scores for financial risks in the pharmaceutical supply chain were 3.2 (SD=0.83) (Supplementary file Table 1).

## Supply chain risks management practices of pharmaceutical manufacturing companies

The study examined 16 supply chain risk mitigation practices in manufacturing companies and the findings were presented in terms of mean scores and standard deviations. In relation to building long-term relationships, 73 respondents (42.4%) agreed that the company values long-term collaborative relationships with its key suppliers, with a mean score of 3.33 (SD=905). Additionally, 87 respondents (50.6%) agreed that the firm and its key suppliers collaborate in sharing risks, with a mean score of 3.52 (SD=862). Eighty six participants (50.0%) agreed that there is a considerable level of trust between the firm and its key suppliers, with a mean score of 3.42 (SD=1.1). On the other hand, the majority (50.6%) disagreed that the company effectively practices continuous supply chain performance audits, such as quality, cost, and delivery, with a mean score of 2.58 (SD=1.020). Sixty nine respondents (40.1%) disagreed that the practice of maintaining inventory only for long-lead time and critical items was effective, with a mean score of 2.91 (SD=0.972). The pharmaceutical manufacturing companies were found to moderately practice maintaining buffer stocks for both raw materials and finished items, with a mean score of 2.95 (SD=1.010). Moreover, 92 respondents (53.5%) agreed that the company considers insurance as a key means of mitigating supply chain risks. The descriptive findings on supply chain risk management (SCRM) practices indicate that, except for building long-term relationships with a mean score of 3.4 (SD=0.78) and risk transfer practices with a mean score of 3.2 (SD=0.78), the majority of the mean scores for other SCRM practices are ≤3 (Table 2 and Supplementary file Table 2).

## Operational performances

The operational performance of the company was evaluated based on cost, quality service, on-time delivery, and flexibility. Seventy nine respondents (45.9%) agreed that the company provides defect-free products, with a mean score of 3.56 (SD=1.03). However, 74 respondents (43%) disagreed that their company has strong strategies in place to minimize costs

**Table 2. Analysis of the SCRM practice in pharmaceutical company of Addis Ababa, Ethiopia, 2023 (N=172).**

| Supply chain risk management practices | Mean | SD |
|---|---|---|
| The company treasures have a collaborative relationship with its key suppliers | 3.33 | .905 |
| The company collaborate with its key suppliers in the areas of sharing risks | 3.52 | .862 |
| There is considerable trust between the company and its key suppliers | 3.42 | 1.119 |
| The supply chain risks for the organization are known and documented | 2.27 | 1.113 |
| In the company, risk management practices are inclusive and participatory | 3.13 | 1.041 |
| The company categorize the supply chain risks as high, medium & low | 3.25 | .992 |
| Risk awareness practices are matured or common in the organization | 2.97 | .988 |
| The company maintains buffer stocks for both raw and finished items | 2.95 | 1.010 |
| In company, inventory is only maintained for long-lead time & critical items | 2.91 | .972 |
| The buffer stocks are maintained considering to minimizing stock holding cost, obsceneness and damage in the company | 3.00 | .979 |
| The company identifies the potential supplier risks reports during vendor appraisals | 3.12 | .990 |
| The company undertakes continuous supply chain performance audits (quality, cost, delivery) | 2.58 | 1.020 |
| The company maintains a backup supplier for pharmaceutical products | 3.04 | .957 |
| The supply chain contingency planning is a critical element of the Business for the company | 3.16 | .941 |
| The company plan contributes to loss minimization, safe assets and risk mitigation | 3.04 | 1.011 |
| Company considers insurance as a key for mitigating supply chain risks | 3.28 | .964 |

related to operations, inventory, warehouse, unnecessary wastage, transportation, and labor, with a mean score of 2.85 (SD=1.101). Furthermore, 79 respondents (45.9%) disagreed that the company has the ability to be flexible enough to adapt to changes in the work environment and demonstrate openness to new ideas in the workplace, with a mean score of 2.90 (SD=1.127). In terms of delivery reliability, 81 respondents (47.1%) disagreed that the company properly executes clients' orders and satisfactorily meets client demand, with a mean score of 2.90 (SD=0.977) (Table 3).

### Regression analysis

**Multiple linear regression assumptions.** Before interpreting the regression analysis, all the assumptions of the multiple regressions should be fulfilled to get the reliable and dependent result of the analysis. Accordingly, normality test, linearity test and multicollinearity test were performed.

**Normality test.** A visual examination of histogram showed a normal distribution of residuals against the predicted dependent variable scores. The absolute value of skewness and kurtosis of the data were tested and found within the accepted threshold value, Z<3.29 ( Table 4 and S1 Fig).

**Linearity test.**

**Homoscedastic test.** To assess linearity, scatter plots of residuals were inspected. As outliers were previously removed, the scatter plots presented below demonstrate a linear relationship between the variables. Homoscedasticity can be evaluated by visually examining a plot of the standardized residuals plotted against the standardized predicted values from the regression. Heteroscedasticity is indicated when the scatter plot is uneven, often exhibiting fan or butterfly patterns, which signify a violation of the homoscedasticity assumption. Additionally, the residuals output have a sound normal distribution because the plotted residuals were around the diagonal straight line (S2 and S3 Fig).

**Multicollinearity.** This assumption can be evaluated by examining the tolerance and the variance inflation factor (VIF). A VIF value below 10 and tolerance statistics above 0.2 indicate the absence of multicollinearity within the data. Based on the data presented the VIF is less than 2, and the tolerance exceeds 0.2. Consequently, it can be concluded that there is no multicollinearity present in the regression model. In addition, the value of correlation coefficient (r) <1 for all bivariate correlation among independent variables which indicates there is no multicollinearity problem in the model (Supplementary file Table 3 and supplementary Table 4).

Table 3. Operational performances in pharmaceutical company of Addis Ababa, Ethiopia, 2023 (N=172).

| Operational performance | Level of agreement | | | | | |
|---|---|---|---|---|---|---|
| | SD (%) | DA (%) | N (%) | A (%) | SA (%) | Mean |
| The company avail the products with the reasonable price | 14(8.1) | 73(42.4) | 25(14.5) | 54(31.4) | 6(3.5) | 2.80 |
| The company have strategies to minimize the cost of operations, inventory and warehouse | 12(7.0) | 74(43.0) | 21(12.2) | 57(33.1) | 8(4.7) | 2.85 |
| The company ensure product quality at each stage | 3(1.7) | 63(36.6) | 17(9.9) | 59(34.3) | 30(17.4) | 3.29 |
| The company provides defect free products for customers | 1(0.6) | 38(22.1) | 25(14.5) | 79(45.9) | 29(16.9) | 3.56 |
| Company identified key processes and activities that influence product quality | 6(3.5) | 81(47.1) | 15(8.7) | 49(28.5) | 21(12.2) | 2.99 |
| The company produce multiple variant products | 4(2.3) | 93(54.1) | 14(8.1) | 51(29.7) | 10(5.8) | 2.83 |
| The company undertake proactive and reactive adaptation of settings to deal with uncertainties | 1(0.6) | 73(42.4) | 34(19.8) | 53(30.8) | 11(6.4) | 3.00 |
| Company is flexible to accommodate the changes in the working environment | 7(4.1) | 79(45.9) | 28(16.3) | 40(23.3) | 18(10.5) | 2.90 |
| The company utilize outsourcing for non-competence activities | 9(5.2) | 71(41.3) | 26(15.1) | 58(33.7) | 8(4.7) | 2.91 |
| Company Properly execute the clients' order | 1(0.6) | 81(47.1) | 31(18.0) | 52(30.2) | 7(4.1) | 2.90 |
| Company satisfies the client's demands for products | 1(0.6) | 81(47.1) | 22(12.8)) | 60(34.9) | 60(34.9) | 2.96 |

**Table 4. Skewness and Kurtosis value for normality test in pharmaceutical companies of Addis Ababa, Ethiopia 2023 (N=172).**

| Parameters | DR | SR | RR | IR | CR | PR | FR | Cost | quality | Flexa* | Delivery |
|---|---|---|---|---|---|---|---|---|---|---|---|
| Skewness | -.725 | -.146 | -.105 | -.106 | .667 | .058 | .083 | .101 | .262 | .254 | .219 |
| Std. Error of Skewness | .185 | .185 | .185 | .185 | .185 | .185 | .185 | .185 | .185 | .185 | .185 |
| Kurtosis | .158 | -.799 | -.667 | -.757 | .044 | -.698 | -.788 | -.683 | -.859 | -.480 | -.694 |
| Std. Error of Kurtosis | .368 | .368 | .368 | .368 | .368 | .368 | .368 | .368 | .368 | .368 | .368 |

**Notice:** DR=Demand SR=Supply RR= Regulatory IR=Infrastructure CR=Catastrophic PR= Production FR= Financial risks & *flexibility

## Multiple regression analysis

**Model summary.** The correlation test was done with multiple linear regressions. The model summary of the regression (R=.546, P<.001) showed the presence of significant correlation between independent and dependent variables indicating model fitness the adjusted R² explains 26.8% of the factor affecting operational performances (Table 5).

## Analysis of ANOVA

The ANOVA result revealed the significance of the regression model which an F significance value of p<0.01. This implies that the regression model has a less than 0.01 likelihood of giving a wrong prediction. Therefore it is concluded that the regression model (with seven independent variables) significantly predicts the outcome variable (Supplementary file Table 5).

## Multiple liner regression Coefficients

A multiple regression analysis was conducted to test the relationship between the independent variables and dependent variable.

$$Y=\beta 0+\beta 1X1+\beta 2X2+\beta 3X3+\beta 4X4+\beta 5X5+\beta 6X6 +\beta 7X 7+\varepsilon$$

**Table 5. Model Summary for good fit of the data Statistics and Coefficients for Risks associated with supply chain performance in pharmaceutical company of Addis Ababa, Ethiopia, 2023 (N=172).**

| R | R² | Adjusted R² | Std. Error of the Estimate | R² Change | F Change | df1 | df2 | Sig. F Change |
|---|---|---|---|---|---|---|---|---|
| .546a | .298 | .268 | .47444 | .298 | 9.935 | 7 | 164 | .000 |
| Model | | Unstandardized Coefficients | | Standardized Coefficients | | | T | Sig. |
| | | B | Std. Error | Beta | | | | |
| (Constant) | | 2.99 | | | | | | |
| Demand side Risk | | -.191 | . 046 | -.309 | | | -4.162 | .011 |
| Supply side Risk | | -.131 | .065 | -.156 | | | -2.015 | .046 |
| Catastrophic risk | | .138 | .041 | .226 | | | 3.335 | .001 |
| Production risk | | -.176 | .058 | -. 216 | | | -3.041 | .003 |
| Regulatory risk | | .095 | .044 | .146 | | | 2.170 | .031 |
| Financial risk | | -.244 | .050 | -.267 | | | -3.811 | .038 |
| Infrastructure risks | | -.322 | .049 | -.250 | | | -3.632 | .005 |

[a]Predictors: (Constant), Demand, Supply, Regulatory,Infrastructure,Catastrophic, Production, Financial risks

Where: Y= operational performance of the firms

β0= Constant (value of y when x1, x2, x3, x4, x5, x6 and x7=0)

β1= Regression coefficient for demand side risk (x1)

β2= Regression coefficient for supply side risk (x2)

β3= Regression coefficient for catastrophic risk (x3)

β4= Regression coefficient for production risk(x4)

β5= Regression coefficient for regulatory risk (x5)

β6= Regression coefficient for financial risk (x6)

β7= Regression coefficient for infrastructure risk (x7)

ε= error term

The Unstandardized coefficients β of the independent variables were substituted in the model (Y= β0+β1X1+β2X2+β3X-3+β4X4+β5X5+ β6X6 + β7X 7+ε). It could be formulated the model as shown below:

$$Y = 2.99 - 0.191X_1 - 0.131X_2 + 0.138X_3 - 0.176X_4 + 0.095X_5 - 0.244X_6 - 0.322X_7 + \varepsilon$$

From the model formula the constant value (β0 = 2.99) implies that organizational performance of pharmaceuticals manufacturing companies in Addis Ababa would be 2.99 if other variables of the model were zero. The regression analysis revealed that demand (β=-0.191, t = -4.162, p < 0.05) and supply side risks (β=-0.131, t = -2.015, p < 0.05) have a negative effect on the operational performance of pharmaceutical manufacturing companies. Holding all other variables constant, a unit increase in demand-side risk leads to a 19.1% decrease in the operational performance of the companies. Similarly, keeping all other variables constant, a unit increases in supply-side risk results in a 13.1% decrease in the operational performance of the manufacturing companies. Finally, hypothesis (H1, H2, H4, H6, H7) were approved they had significant negative effect on the operational performance of the pharmaceutical manufacturing companies (Table-5).

## Qualitative findings

The qualitative part of the study involved conducting face-to-face interviews with key informants. The participants included key informants between the ages of 30 and 41 who had a minimum of 5 years of experience in pharmaceutical manufacturing companies. From the data collected, four main themes emerged: costs, natural and man-made risks, logistics-related risks, and methods for managing supply chain risks and their constraints. The findings can be summarized as follows.

## Cost/currency related issues

The pharmaceutical supply chain faces various risks that can disrupt the supply of pharmaceuticals and supplies. Managing these risks is crucial for improving the company's performance. Many company managers expressed concerns about the increasing costs of freight and limitations related to currency, particularly the fluctuating foreign exchange rates for importing raw materials. These risks significantly impact the company's operations.

One of the supply chain managers with six years of working experiences stressed the challenges faced by the company as follows, *"...our company encounters numerous supply chain risks in our daily activities. As you know, all our active ingredients and excipients are imported from China, India and others countries which need foreign currency. However, we often experience shortages of raw materials due to limited availability of hard currency permitted by banks. Additionally, foreign exchange rates fluctuation and cost of freight is continuously rising which further exacerbate the shortage of foreign currency. There are many factors to consider when assessing risks in the pharmaceutical supply chain. For instance, the increase in oil prices has had a significant impact on our supply chain operations."* The production manager also expressed concerns, stating, *"...over the past three years, we have faced shortages of supplies due to foreign currency*

*limitations. Many of our production lines have experienced decreased capacity, and some have even been forced to shut down. Delays in supplies have resulted in shortages of raw materials and reduced shelf life."* Overall, the financial and currency-related risks in the pharmaceutical supply chain pose significant challenges for companies, impacting their operations and supply of essential pharmaceutical products.

## Natural and man-made supply chain risks

It is evident that catastrophic risks are a significant concern for the majority of participants. The causes of vulnerabilities in the supply chain include uncertainty and emergency conditions such as epidemics, pandemics, political instabilities, social conflicts, and economic and socio-economic growth. These factors have a profound impact on the daily activities of pharmaceutical manufacturing companies. It is crucial for companies to have pre-planned activities in place to effectively manage such conditions when they occur. Key informants suggested that in order to meet customer demand and address supply chain disruptions caused by emerging pandemics, currency shortages, and other pharmaceutical supply chain risks, companies should maintain safety stock for long shelf-life products. During the interviews, participants mentioned that companies have been exposed to various factors that disrupt the supply chain, including conflicts and the COVID-19 pandemic. One key respondent elaborated on the problem, stating, *"...we have observed that pharmaceutical supply chain risks have led to shortages of supplies. The COVID-19 pandemic and internal conflicts have resulted in decreased production capacity, increased overall costs, and limited availability of pharmaceuticals. Additionally, there is the issue of forecast error, particularly when dealing with long lead times. The forecast error tends to multiply exponentially as the lead time extends. It's like trying to forecast the weather tomorrow versus next month; you have no idea what will happen next month. Similarly, in this situation, there is a significant risk of incorrect forecasting, which occurs repeatedly. As a result, the company experiences drastic fluctuations between shortages and excess inventory."* In generally, factors such as uncertainty, emergency conditions, and forecast error contribute to supply chain vulnerabilities. Companies should proactively plan and manage these risks to ensure the availability of pharmaceuticals and mitigate disruptions in the supply chain.

## Logistics related issues

Manufacturing companies face on-going challenges in managing inventory and ensuring successful product delivery to customers. However, the key informants (KIs) mentioned that their lack of logistics expertise and limited options for logistics service providers have hindered their performance. One of the respondents, a supply chain manager, explained, *"Unfortunately, we have a limited number of logistics service providers that we rely on, which makes it more challenging to manage and mitigate risks. The lack of flexibility in decision-making regarding logistics services has raised costs. Currently, all logistics activities are under government control."*

## Supply chain risks management methods and constraints

Regarding supply chain risk management (SCRM) methods and constraints, participants identified five primary approaches. These approaches include coordinating and collaborating with supply chain partners, seeking alternative local suppliers, maintaining a shortlist of trusted suppliers, purchasing cargo insurance, and evaluating and discussing risks and mitigation methods with staff. However, there is a lack of pre-planned activities specifically designed for supply chain risks. The absence of an organized risk mitigation team in the company has made it more challenging to address supply chain risks. Most KIs stated that there is no well-documented and regularly practiced SCRM mitigation method in the company, although various methods exist at a cost. They suggested that the company needs to identify and assess different types of risks, implement formally documented SCRM practices, strengthen its supply chain units with technology, enhance coordination and collaboration with stakeholders, and create awareness of supply chain risks and SCRM among employees. The supply chain manager further elaborated on the problem, saying, *"Even though our company*

*practices some supply chain management (SCM) techniques based on experience and exposure, we lack a previously developed framework for risk identification and management, as well as a risk monitoring system. This may be due to a lack of awareness among employees regarding the risks associated with the supply chain. Again, there are various ways to mitigate these risks, but they come at a cost."*

Another participant shared their perspective, stating, *"...in our organization, we lack an organized team and framework specifically designed for risk reduction in emergency conditions. We attempt to identify and discuss issues within our department and share the situation with the responsible body to address and resolve the problems."* (This comment was made by research development manager with seven years of work experience).

## Discussion

The objective of the study was to analyze how supply chain risk management practices affect the performance of pharmaceutical manufacturing companies. Risks within the supply chain can negatively affect the healthcare system's efficiency and disrupt the delivery of pharmaceuticals. By possessing sufficient knowledge of these risk management practices, supply chain organizations can reduce costs, minimize waste, mitigate liabilities, and enhance overall supply chain performance [36].

The study found that demand side supply chain risks were a concern for the pharmaceutical manufacturing companies surveyed. Of the respondents, 84 (48.8%) and 79 (45.9%) acknowledged the risks of customers' unanticipated demand and insufficient or distorted information, respectively. These findings are comparable to those reported by the Ethiopian pharmaceutical supply service, which faced major risks of 54.2% due to unexpected increases in demand and information distortion [24]. The results are also somewhat consistent with a study conducted in the UK, where demand side risks accounted for 33.2% [37]. Similarly, a study in the Indian pharmaceutical industry identified incorrect forecasting, poor quality information sharing among stakeholders, and inability to meet unexpected surge in demand due to capacity constraints as key demand side risks [26]. The key informants in the current study emphasized that unacceptable forecast errors, long lead times, unexpected surge of demands, and lack of quality data were the major issues related to demand side risks.

The pharmaceutical manufacturing companies assessed identified supplier side risks as a significant threat, with concerns surrounding some suppliers, lead times, distribution risks, and item unavailability [25]. Of the respondents, 87 (50.6%), 82 (47.7%), and 79 (45.9%) agreed that poor logistic service performance of suppliers, company dependency on key suppliers, and fluctuations or shortages in supply markets, respectively, were key supplier side risks. These findings are consistent with a study conducted at the Ethiopian pharmaceutical supply service, which found that 48.6% of supply chain risks were associated with poor supplier performance [24]. However, studies conducted in Iran and the UK reported lower percentages of supplier side risks at 40% and 34.6%, respectively [14,37]. The difference may be attributed to variations in study methodology and country context. Key informants in the current study emphasized that deprived logistics performance of suppliers, reliance on only key suppliers, scant performance of logistics service providers, inadequate monitoring of suppliers, increases in product prices by suppliers, and capacity fluctuations in the supply markets were major bottlenecks for pharmaceutical manufacturing companies in Ethiopia.

In terms of financial risks in the supply chain of pharmaceutical manufacturers, a majority of respondents (59.3%, 45.93%, and 48.26%) identified high freight charges, financial restrictions on product availability, and dynamic foreign exchange rates as significant threats to pharmaceutical manufacturing companies. A study conducted in Iran found that the majority of supply chain risks were related to financial and economic factors such as currency fluctuation (90.18%), inflation rates (76.64%), and pricing policies (70.07%) [29]. Similarly, a study conducted at EPSA's highlighted the struggles of the supply chain with currency fluctuations, high freight costs, and a lack of hard currency [24]. Key informants in the current study also noted that pharmaceutical active ingredients and excipients are import-based, leading to difficulties in importing necessary items on time due to limited hard currency, constant foreign exchange rate fluctuations, and increased freight costs.

Concerning risks related to pharmaceutical manufacturing infrastructure, 37.8% of respondents admitted internal IT infrastructure and system breakdowns, while 29.1% reported external IT infrastructure interruptions. This poses more significant challenges when compared to similar studies conducted in the United Kingdom and the United States [38,39]. Additionally, a report on Kenya's electricity industry highlights supply chain disruptions, including stock outages, fire outbreaks, IT system malfunctions, and environmental disturbances [30], indicating suboptimal information sharing with suppliers and customers that could lead to information distortions. The key informant responses revealed infrastructure risks such as breakdowns in internal and external IT infrastructure, downtime or loss of production capacity due to local disruptions, infrastructure unavailability, and supply disruptions within their respective firms.

Addressing pharmaceutical manufacturing risks is crucial for improving the overall performance and efficiency of pharmaceutical supply chains, ensuring the availability of quality medications, and meeting the healthcare needs of the population. Implementing robust risk management strategies, investing in IT infrastructure, enhancing supplier relationships, and fostering collaboration among stakeholders can help mitigate these risks and strengthen the pharmaceutical supply chain [40,41]. The emerging integration of artificial intelligence (AI) in pharmaceutical supply chain management offers a promising solution for mitigating supply chain risks within companies. However, the adoption of this technology faces basic challenges related to IT infrastructure and the availability of skilled human resources especially in low income countries. For example, AI has the ability to handle and analyze large volumes of data, facilitating proactive risk identification and offering early management suggestions. By enhancing decision-making processes, improving supply chain visibility, and boosting the resilience and operational efficiency of the company, AI technology proves instrumental in optimizing pharmaceutical supply chain operations [42,43].

However, in the current study among study participants only about one-fourth 50.6%, agreed that the firm and its key suppliers collaborate in sharing risks and 45.4% respondents agreed that there were constant internet interruption internal organization and external organizations. These challenges hinder the initiation and implementation of data visibility and AI technologies in supply chain risks management of manufacturing companies within developing nations like Ethiopia. The key informants also suggested that even though managing supply chain risks through coordination and collaboration among supply chain partners is necessary to ensure profitability and continuity. The most pharmaceutical manufacturer companies were lacks of identifying and document the supply risks, framework for risk identification and management, continuous supply chain performance audits, collaboration with its key suppliers in the areas of sharing risks, supply chain contingency planning, employee adequate awareness about supply chain risks and its mitigation practice. The systematic reviews conducted India also revealed that supplier and financial risk sides are major factors that influence the operation performance of manufacturing companies [14]. The similar study conducted in Brazilian pharmaceutical companies identified regulator and financial side risk are the major that affect pharmaceutical manufacturing companies[16]. Again, these issues were the similar to the studies conducted in Greek pharmaceutical companies and Australia pharmaceutical industries and Brazilian pharmaceutical companies[39,40,44].

The pharmaceutical manufacturing companies should measure their operational performance using cost, service quality, delivery time, and flexibility/responsiveness dimension. However, in the current study 43% of respondents disagreed with the presence of strong strategies to minimize the cost of operations, inventory, and warehousing. According to key informants, the supply of global medicines is unreliable during times of crisis in various parts of the country, and the COVID-19 pandemic has resulted in drug shortages and significant price increases. Limited availability of hard currency and hard currency inflation has also contributed to the increased prices of pharmaceuticals and overall logistic service costs. A similar study conducted in Iran demonstrated that the cost of operations greatly affects profitability and is an important indicator for evaluating efficiency [45]. There is strong suggestion that the company need to give attention to emergency preparedness and flexibility issues to give quick responses during emergency time [1,2]. However, of the respondents 45.9% and 47.1% disagreed about their pharmaceutical manufacturing companies' flexibility and responsive, respectively.

A summary of the multiple linear regression model (R=.546, P< 001) indicates a significant correlation between the independent and dependent variables. An R² value of 26.8% means that supply chain risks can explain approximately 26.8% of the variation in organizational performance. The regression analysis further reveals that demand (β=-0.191, t = -4.162, p < 0.05) and supply (β=-0.131, t = -2.015, p < 0.05) side risks have a negative impact on the operational performance of pharmaceutical manufacturing companies, with a unit increase in demand-side risk causing to a 19.1% decrease and supply-side risk resulting in a 13.1% decrease in operational performance. Additionally, financial (β= 0.244, t = -3.811, p < 0.05) and infrastructure (β= 0.322, t = --3.632, p < 0.05) related supply risks also have a negative effect, with a unit increase in financial-side risk leading to a 24.4% decrease and infrastructure-side risk causing in a 32.2% decrease in operational performance. This study finding are share the similar views with the study conducted on adoption of green supply chain initiatives in the pharmaceutical industry in the India where the Infrastructure risks, with a weight score of 0.40, are ranked as the primary concern. Following closely, the supply side risks emerge as the second most influential factors affecting operational performance, with the financial risk category being ranked fifth [46]. Furthermore, the studies conducted in Iran and Ghana revealed that regulatory and financial side risks significantly influences the operational performance of manufacturing companies [29,47].

**The study strength, limitation and future research recommendation:** The current study included quantitative and qualitative findings that support each other to obtain more detail information about pharmaceutical supply chain risks on the operational performance. Less similar primary research was accessible to approve the proposed framework, and we used a few non-health articles to develop the conceptual framework. This limitation restricts our capacity to discuss and explain the study findings in detail. Furthermore, it posed challenges when comparing the risks of pharmaceutical supply chain with non-pharmaceutical. Only the staffs working in pharmaceutical manufacturing companies found in Addis Ababa, Ethiopia were included due to time, budget, and human resource constraints. This study is limited to pharmaceutical manufacturing companies in Ethiopia and requires additional research on supply chain risks concerning suppliers and health facilities, including importers, wholesalers, and service delivery points in health facilities. The nature and scope of risks may differ, and their management strategies vary, as supply chain management encompasses the entire pipeline, not just manufacturers.

## Conclusion

The research study revealed supply chain risks faced by pharmaceutical manufacturing firms, including inaccurate forecasting, lengthy lead times, unforeseen spikes in demand, insufficient data quality, poor logistics performance by suppliers, inadequate supplier monitoring, fluctuations in manufacturing capacity, limited access to hard currency, fluctuations in foreign exchange rates, rising freight costs, disruptions in internal and external IT systems, failure to identify and document supply risks, lack of a risk identification and management framework, limited collaboration with key suppliers to share risks, and inadequate employee awareness of supply chain risks and mitigation strategies. Demand, supply, financial and infrastructure related supply chain risks had significant negative influence on operational performance of pharmaceutical manufacturing companies. The operational performance of pharmaceutical manufacturing companies is negatively affected by various supply chain risks, including those related to demand, supply, finance, and infrastructure. So, companies should give due attention for those risks and have management framework to decrease the consequences of those threats and increase their operation performances.

Furthermore, the Federal Ministry of Health and Industry should closely monitor the implementation of SCRM practices in Addis Ababa's pharmaceutical company. This can be achieved by minimizing budget and currency issues and developing policies and procedures that positively impact the company's operations.

## Supporting Information

**S1 Table. Supplementary file; Supply chain risks in pharmaceutical companies of Addis Ababa, Ethiopia, 2023 (N=172).**
(ZIP)

**S2 Table.  Supplementary file; Supply chain Risk mitigation practices in pharmaceutical firms of Addis Ababa, Ethiopia, 2023(N=172).**
(ZIP)

**S3 Table.  Supplementary file; The result of multicollinearity test for independent variables in pharmaceutical companies of Addis Ababa, Ethiopia, 2023 (n=172).**
(ZIP)

**S4 Table.  Supplementary file; Bivariate correlation among the independent and dependent variable in pharmaceutical companies of Addis Ababa, Ethiopia, 2023 (n=172).**
(ZIP)

**S5 Table.  Supplementary file; Significance level for multiple correlation coefficient-ANOVA in pharmaceutical companies of Addis Ababa, Ethiopia, 2023 (N=172).**
(ZIP)

**S6 Table.  Supplementary file survey questioner annex.**
(ZIP)

**S1 Fig.  Supplementary file; Histogram on a normal distribution of residuals against the predicted dependent variable scores in pharmaceutical companies of Addis Ababa, Ethiopia, 2023(N=172).**
(ZIP)

**S2 Fig.  Supplementary file; Scatter Plot Based on Residual and predictive value in pharmaceutical companies of Addis Ababa, Ethiopia 2023(N=172).**
(ZIP)

**S3 Fig.  Supplementary file; P-P Plot of Standardized Regression Standardized Residual in pharmaceutical companies of Addis Ababa, Ethiopia 2023 (N=172).**
(ZIP)

## Acknowledgments

We would like to thank Jimma University for facilitating the study and covering stationery materials, staffs of pharmaceutical manufacturing companies in Addis Ababa for cooperating during data abstraction. We would also like to acknowledge all data collectors and supervisors respondents without whom this research would not have been realized.

## Author contributions

**Conceptualization:** Bekele Boche.

**Data curation:** Bekele Boche, Dinku Mechal.

**Formal analysis:** Dinku Mechal.

**Methodology:** Bekele Boche, Dinku Mechal.

**Software:** Dinku Mechal.

**Writing – original draft:** Bekele Boche, Dinku Mechal.

**Writing – review & editing:** Bekele Boche, Dinku Mechal.

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
