## [Decision Letter · Decision Letter 0]

28 Jan 2025

PONE-D-24-50278The effect of supply chain risks management practices on operational performance of pharmaceutical manufacturing companies in Addis Ababa, Ethiopia:Analytical cross-sectional studyPLOS ONE

Dear Dr. Boche,

Thank you for submitting your manuscript to PLOS ONE. After careful consideration, we feel that it has merit but does not fully meet PLOS ONE’s publication criteria as it currently stands. Therefore, we invite you to submit a revised version of the manuscript that addresses the points raised during the review process.

We look forward to receiving your revised manuscript.

Kind regards,

João Zambujal-Oliveira

Academic Editor

PLOS ONE

Journal Requirements:

3. Thank you for stating the following in your Competing Interests section:  The authors declare that they have no known competing financial interests or personal relationships that could have appeared to influence the work reported in this paper.

Reviewers' comments:

Reviewer's Responses to Questions

**Comments to the Author**

1. Is the manuscript technically sound, and do the data support the conclusions?

Reviewer #1: Yes

Reviewer #2: Yes

2. Has the statistical analysis been performed appropriately and rigorously? 

Reviewer #1: Yes

Reviewer #2: Yes

3. Have the authors made all data underlying the findings in their manuscript fully available?

Reviewer #1: Yes

Reviewer #2: Yes

4. Is the manuscript presented in an intelligible fashion and written in standard English?

Reviewer #1: Yes

Reviewer #2: Yes

5. Review Comments to the Author

Reviewer #1: Interesting work and findings as it provides insight of supply chain risk management in a geographical context that is less explored.

Abstract is clear summarizing the work but can be improved with highlights on how these findings contribute to global supply chain risk management field and policy/industry practices.

For literature review, please conduct comparison and critically analyse the similar studies in other countries/industries to provide more context and shows the needs of this research.

Methods are clear but it should explain more on limitations acknowledged in the paper and how this impacted the findings.

The findings of this study should be relate or associated with current or existing supply chain risk management theories or model in order to show the significant of the results to the SCRM field.

Discussion on how the findings could be relevant to other developing countries or even to global supply chain facing similar risks.

From the findings of this risk, authors can discuss potential application of technology such as AI to mitigate the risks.

Reviewer #2: An article of high substantive level and with an appropriate literature review. Alternatively, one could consider a more readable presentation of the issue of assessing the reliability of the research tool (survey questionnaire) in the form of a table. One could also consider the justification for using so many statistical methods in verifying research hypotheses. I propose to indicate more clearly the direction of future research in the undertaken research area.

6. PLOS authors have the option to publish the peer review history of their article (what does this mean? ). If published, this will include your full peer review and any attached files.

**Do you want your identity to be public for this peer review?** For information about this choice, including consent withdrawal, please see our Privacy Policy .

Reviewer #1: No

Reviewer #2: No

---

## [Author Response · Author response to Decision Letter 1]

19 Feb 2025

Esteemed editor and reviewers, we extend our sincere gratitude for your invaluable and constructive suggestions and comments. Your input has been carefully considered, and we have made effort to address your concerns as per your guidance, incorporating them into the rebuttal letter and the manuscript documents.

Raised comments suggestion and the proposed response

Response: Thank you for reminding us and for your suggestion. We have carefully reviewed the writing style you mentioned and have made adjustments to align with it.

Response: We have verified and authenticated the corresponding author ORCID in the PLOS ONE. ORCID: https://orcid.org/0000-0002-9218-9669

3. Thank you for stating the following in your Competing Interests section: The authors declare that they have no known competing financial interests or personal relationships that could have appeared to influence the work reported in this paper. Please complete your Competing Interests on the online submission form to state any Competing Interests. If you have no competing interests, please state ""The authors have declared that no competing interests exist."", as detailed online in our guide for authors at http://journals.plos.org/plosone/s/submit-now

Response: We have addressed any potential competing interests. The statement "The authors have declared that no competing interests exist" was submitted through manuscript document and cover latter pre your respond to my Gmail request response “Thank you for reaching out.

If the competing interest statement is already included within your paper you may now resubmit (as it is). I will gladly updated the online submission on your behalf”

4. Please amend the title either on the online submission form or in your so that they are identical

Response: Thank you for reminding us to verify the consistency between the online submission and the manuscript document. We have reviewed and corrected it as per your comments.

5. We note that you have indicated that there are restrictions to data sharing for this study. For studies involving human research participant data or other sensitive data, we encourage authors to share de-identified or anonymized data. However, when data cannot be publicly shared for ethical reasons, we allow authors to make their data sets available upon request. Before we proceed with your manuscript, please address the following prompts:

Response: All relevant data are within the manuscript and its Supporting Information files. The data is accessible from the authors and can be shared with no ethical or legal restrictions upon request. The data were collected and handled by the authors. So, it is not owned by a third-party organization.

b) If there are no restrictions, please upload the minimal anonymized data set necessary to replicate your study findings to a stable, public repository and provide us with the relevant URLs, DOIs, or accession numbers. Please see http://www.bmj.com/content/340/bmj.c181.long for guidelines on how to de-identify and prepare clinical data for publication. For a list of recommended repositories, please see https://journals.plos.org/plosone/s/recommended-repositories. You also have the option of uploading the data as Supporting Information files, but we would recommend depositing data directly to a data repository if possible. Please update your Data Availability statement in the submission form accordingly.

Response: We have revised our statement on data availability and sharing in both the submission platform and the manuscript document per your recommendation.

6. Please include captions for your Supporting Information files at the end of your manuscript, and update any in-text citations to match accordingly.

Response: Thank you for your suggestion. We have incorporated the captions for the supporting information files at the end of the manuscript document as per your recommendation.

Response: Thank you for your suggestions. We have thoroughly reviewed all references and did not come across the retracted reference.

Reviewers' comments:

Reviewer's Responses to Questions

Comments to the Author

1. Is the manuscript technically sound, and do the data support the conclusions?

Reviewer #1: Yes

Reviewer #2: Yes

Response: Thank you for providing your review. I sincerely appreciate your understanding and encouragement.

2. Has the statistical analysis been performed appropriately and rigorously?

Reviewer #1: Yes

Reviewer #2: Yes

Response: Thanks

3. Have the authors made all data underlying the findings in their manuscript fully available?

Reviewer #1: Yes

Reviewer #2: Yes

Response: Thanks

4. Is the manuscript presented in an intelligible fashion and written in standard English?

Reviewer #1: Yes

Reviewer #2: Yes

Response: Thanks

5. Review Comments to the Author

Reviewer #1:

1. Interesting work and findings as it provides insight of supply chain risk management in a geographical context that is less explored. Abstract is clear summarizing the work but can be improved with highlights on how these findings contribute to global supply chain risk management field and policy/industry practices.

Response: Thank you for your suggestion. We have integrated your concern into the abstract, conclusion section, and emphasized it.

2. For literature review, please conduct comparison and critically analyse the similar studies in other countries/industries to provide more context and shows the needs of this research. Response: Thank you for your input. We have endeavored to incorporate your recommendation by reviewing study findings from various countries or industries.

3. Methods are clear but it should explain more on limitations acknowledged in the paper and how this impacted the findings.

Response: We have detailed the limitations of the study and discussed how they impact the results of our research.

4. The findings of this study should be relate or associated with current or existing supply chain risk management theories or model in order to show the significant of the results to the SCRM field. Discussion on how the findings could be relevant to other developing countries or even to global supply chain facing similar risks. From the findings of this risk, authors can discuss potential application of technology such as AI to mitigate the risks.

Response: Thank you for your valuable input and suggestions, particularly regarding AI. We have made an effort to address your points by incorporating your suggestions into the main document. Dear reviewer, following our research, we have explored studies conducted in developing countries. In light of limitations observed in previous studies within this domain, we found and included developing nations like Ghana and Ethiopia. Additionally, we have considered the study of the electricity industry supply chain in Kenya; however, it differed from the pharmaceutical products supply chain. Further we have enhanced our findings by leveraging insights from developed and middle-income countries.

Reviewer #2:

5. An article of high substantive level and with an appropriate literature review. Alternatively, one could consider a more readable presentation of the issue of assessing the reliability of the research tool (survey questionnaire) in the form of a table. One could also consider the justification for using so many statistical methods in verifying research hypotheses. I propose to indicate more clearly the direction of future research in the undertaken research area.

Response: Thank you for your feedback and suggestions. The survey questionnaire used has been included in the supplementary file for reference. A potential future research recommendation was written under the section titled "Research Strengths and Limitations."

Thank you immensely for your valuable input!

---

## [Decision Letter · Decision Letter 1]

4 Mar 2025

The effect of supply chain risks management practices on operational performance of pharmaceutical manufacturing companies in Addis Ababa, Ethiopia:Analytical cross-sectional study

PONE-D-24-50278R1

Dear Dr. Boche,

We’re pleased to inform you that your manuscript has been judged scientifically suitable for publication and will be formally accepted for publication once it meets all outstanding technical requirements.

Kind regards,

João Zambujal-Oliveira

Academic Editor

PLOS ONE

Additional Editor Comments (optional):

Reviewers' comments:

Reviewer's Responses to Questions

**Comments to the Author**

1. If the authors have adequately addressed your comments raised in a previous round of review and you feel that this manuscript is now acceptable for publication, you may indicate that here to bypass the “Comments to the Author” section, enter your conflict of interest statement in the “Confidential to Editor” section, and submit your "Accept" recommendation.

Reviewer #1: (No Response)

Reviewer #2: All comments have been addressed

2. Is the manuscript technically sound, and do the data support the conclusions?

Reviewer #1: (No Response)

Reviewer #2: Yes

3. Has the statistical analysis been performed appropriately and rigorously? 

Reviewer #1: (No Response)

Reviewer #2: Yes

4. Have the authors made all data underlying the findings in their manuscript fully available?

Reviewer #1: (No Response)

Reviewer #2: Yes

5. Is the manuscript presented in an intelligible fashion and written in standard English?

Reviewer #1: (No Response)

Reviewer #2: Yes

6. Review Comments to the Author

Reviewer #1: (No Response)

Reviewer #2: I rate the revised article as very good. The changes made are appropriate and as suggested. I recommend the article for publication in the journal.

7. PLOS authors have the option to publish the peer review history of their article (what does this mean? ). If published, this will include your full peer review and any attached files.

**Do you want your identity to be public for this peer review?** For information about this choice, including consent withdrawal, please see our Privacy Policy .

Reviewer #1: No

Reviewer #2: No

---

## [Editor Report · Acceptance letter]

PONE-D-24-50278R1

PLOS ONE

Dear Dr. Boche,

I'm pleased to inform you that your manuscript has been deemed suitable for publication in PLOS ONE. Congratulations! Your manuscript is now being handed over to our production team.

Kind regards,

on behalf of

Prof. João Zambujal-Oliveira

Academic Editor

PLOS ONE